# Epidemiology of hepatitis B virus co-infection among people living with HIV in four countries in sub-Saharan Africa

Josphat Kosgei[1,2,3]*, Walter Jaoko[3], Julius Oyugi[3], Natalie Burns[4,5], Jonah Maswai[1,2], John Owuoth[6,7], Valentine Sing'oei[6,7], Emmanuel Bahemana[8,9], Abdulwasiu B. Tiamiyu[10,11], Hannah Kibuuka[12], Leigh Anne Eller[4,5], Julie A. Ake[4], Neha Shah[4], Trevor A. Crowell[4,5]*, on behalf of the African Cohort Study (AFRICOS) Study Group[¶]

1 U.S. Military HIV Research Program, CIDR, Walter Reed Army Institute of Research—Africa, Kericho, Kenya, 2 HJF Medical Research International, Kericho, Kenya, 3 College of Health Science, University of Nairobi, Nairobi, Kenya, 4 U.S. Military HIV Research Program, CIDR, Walter Reed Army Institute of Research, Silver Spring, Maryland United States of America, 5 Henry M. Jackson Foundation for the Advancement of Military Medicine, Bethesda, Maryland, United States of America, 6 U.S. Military HIV Research Program, CIDR, Walter Reed Army Institute of Research—Africa, Kisumu, Kenya, 7 HJF Medical Research International, Kisumu, Kenya, 8 U.S. Military HIV Research Program, CIDR, Walter Reed Army Institute of Research—Africa, Mbeya, Tanzania, 9 HJF Medical Research International, Mbeya, Tanzania, 10 U.S. Military HIV Research Program, CIDR, Walter Reed Army Institute of Research—Africa, Abuja, Nigeria, 11 HJF Medical Research International, Abuja, Nigeria, 12 Makerere University Walter Reed Program, Kampala, Uganda

¶ Membership of African Cohort Study (AFRICOS) Study Group is provided in the Acknowledgments.
* josphat.kosgei@usamru-k.org (JK); tcrowell@hivresearch.org (TAC)

## Abstract

### Background

Hepatitis B virus (HBV) and human immunodeficiency virus (HIV) share transmission routes and are associated with morbidity in sub-Saharan Africa. Understanding the prevalence and demographic predictors of HBV co-infection among people living with HIV (PLWH) can inform integrated screening and treatment strategies.

### Methods

We analyzed enrollment data from the African Cohort Study (AFRICOS), a prospective study of participants aged ≥15 years with and without HIV across 12 clinical sites in Kenya, Uganda, Tanzania, and Nigeria. HBV was diagnosed via hepatitis B surface antigen (Bio-Rad, Hercules, CA). Descriptive statistics summarized HBV prevalence by HIV status and demographic characteristics. Relative risk (RR) of HBV among PLWH compared to people without HIV was calculated. Among PLWH, multivariable logistic regression identified odds ratios (ORs) and 95% confidence intervals (CIs) for factors independently associated with HBV co-infection.

**Data availability statement:** The datasets generated and/or analyzed during the current study are not publicly available due to ethical restrictions. A minimal dataset sufficient to reproduce the reported findings is available upon reasonable request from the Data Coordinating and Analysis Center (DCAC) at PubRequest@hivresearch.org, referencing the AFRICOS (RV329) study.

**Funding:** This work was supported by the President's Emergency Plan for AIDS Relief (PEPFAR) and the U.S. National Institutes of Health via agreements between the Henry M. Jackson Foundation for the Advancement of Military Medicine, Inc., and the U.S. Department of Defense (#HT9425-24-2-0020, #W81XWH-11-2-0174, #W81XWH-18-2-0040, and #HT9425-24-3-0004).

**Competing interests:** The authors have declared that no competing interests exist.

**Abbreviations:** AFRICOS, African Cohort Study; ALT, Alanine aminotransferase; ART, Antiretroviral therapy; AST, Aspartate aminotransferase; CI, Confidence interval; DILI, Drug-induced liver injury; ELISA, Enzyme-linked immunosorbent assay; HBsAg, Hepatitis B surface antigen; HBV, Hepatitis B virus; HIV, Human immunodeficiency virus; IQR, Interquartile range; IRB, Institutional Review Board; KEMRI, Kenya Medical Research Institute; NIMR, National Institute for Medical Research; NHREC, National Health Research Ethics Committee; OR, Odds ratio; PEPFAR, President's Emergency Plan for AIDS Relief; PLWH, People living with HIV; RR, Relative risk; TDF, Tenofovir disoproxil fumarate; ULN, Upper limit of normal; UNCST, Uganda National Council for Science and Technology; WRAIR, Walter Reed Army Institute of Research.

## Results

Among 4,241 participants with valid HBV results, 3,439 were PLWH. HBV prevalence was 4.7% among PLWH and 4.2% among people without HIV (RR: 1.10; 95% CI: 0.77–1.59). Among PLWH, HBV prevalence was highest in Nigeria and lowest in Kenya (11.1% vs. 2.6%, $p < 0.001$), higher among males than females (6.7% vs. 3.2%, $p < 0.001$, and highest in participants aged ≥40 years and lowest in participants aged 15–19 years (5.8% vs. 0.2%, $p < 0.001$). In multivariable analysis, male sex (OR: 2.22 [95%CI: 1.59–3.09]), Nigeria residence (OR: 5.19 [95%CI: 3.35–8.05]), and age ≥ 40 (OR: 2.99 [95%CI: 1.74–5.13]) were independently associated with HBV co-infection.

## Conclusion

Routine HBV screening and targeted interventions are needed within HIV programs, especially for older males and in high-prevalence settings like Nigeria.

---

## Introduction

Hepatitis B virus (HBV) and human immunodeficiency virus (HIV) are two major global health concerns that frequently co-occur due to shared transmission routes, including condomless sexual contact, perinatal exposure, and parenteral routes such as injection drug use and unsafe medical practices. Co-infection with HBV and HIV presents significant clinical challenges, as it is associated with more rapid progression of liver disease, higher rates of chronic HBV infection, and increased liver-related morbidity and mortality compared to mono-infection with either virus [1]. Beyond liver disease, HBV co-infection among PLWH has also been associated with higher rates of hospitalization due to non-AIDS-defining infections, including bacterial and non-specific inflammatory conditions, underscoring the broader systemic impact of co-infection [2].

Sub-Saharan Africa bears a substantial burden of both HIV and HBV, with overlapping endemicity in many countries [3]. Although antiretroviral therapy (ART) regimens that are active against both viruses have improved the prognosis of co-infected individuals, HBV remains a major contributor to liver-related complications among people living with HIV (PLWH), particularly in resource-limited settings where HBV screening and vaccination coverage remain low [4].

Understanding the epidemiologic characteristics of HBV among PLWH is critical for informing integrated prevention, diagnosis, and treatment strategies. Previous studies have reported variable prevalence of HBV among PLWH across Sub-Saharan Africa, with co-infection influenced by geographic location, age, sex, and behavioral or healthcare-related exposures [5]. However, large-scale comparative studies evaluating HBV prevalence in people with and without HIV in this region remain limited.

We assessed the prevalence of HBV among a large cohort of individuals from four Sub-Saharan African countries—Kenya, Uganda, Tanzania, and Nigeria—with a

particular focus on PLWH. We further examined demographic and regional associations of HBV co-infection among PLWH using multivariable logistic regression. Our findings contribute to the growing evidence base needed to guide public health policies and clinical management strategies for HIV/HBV co-infected populations in Sub-Saharan Africa.

## Methods

We analyzed prevalence and socio-demographic factors associated with hepatitis B co-infection among PLWH in the African Cohort Study (AFRICOS). AFRICOS is an open-ended prospective cohort study, that enrolled adults and adolescents aged 15 years and above, with and without HIV, at clinical sites in Kenya, Tanzania, Uganda, and Nigeria that are supported by the U.S. President's Emergency Plan for AIDS Relief (PEPFAR) via the U.S. Military HIV Research Program (MHRP). Participants were eligible if they were aged ≥15 years, enrolled in the AFRICOS cohort, and had available HIV and hepatitis B serologic data. Participants without complete HBV serologic results at any visit were excluded from the analysis. No formal sample size calculation was performed, as AFRICOS is an ongoing cohort study. The analytic sample size was determined by available data. The large cohort provides sufficient precision for prevalence estimates and regression analyses.

HBV was diagnosed by the presence of hepatitis B surface antigen detected using an enzyme-linked immunosorbent assay (ELISA) (Bio-Rad, Hercules, CA). Participants with a valid HBV test at any study visit were included in the analytic population using data from the time of their first valid HBV test. Demographic variables included age, sex, and country. Participants were categorized into age groups (15–19 years; 20–24 years; 25–39 years; 40 years and above).

Descriptive statistics were used to summarize HBV prevalence by HIV status and to characterize the demographic distribution of the analytic population. Categorical variables, including HIV and HBV status, sex, country of origin, and age group, were summarized using frequencies and percentages. HBV prevalence was calculated as the proportion of individuals testing positive for HBV among participants with a valid test, stratified by HIV status.

HBV prevalence was compared between people with and without HIV using a chi-square test, and a relative risk (RR) was calculated with 95% Wald confidence interval (CI).

Among PLWH, prevalence of HBV co-infection by age group, sex, and country were compared using chi-square tests or Fisher's exact tests (for variables with any cell size <5). A p-value of <0.05 was considered statistically significant.

To identify factors independently associated with HBV co-infection among PLWH, a multivariable logistic regression model was constructed. The outcome variable was HBV co-infection (present vs. absent), and independent variables were decided a priori to include sex, country, and age group. Unadjusted and adjusted odds ratios (ORs) with 95% CIs were reported for each independent variable, along with associated p-values. All statistical analyses were conducted using R version 4.3.2 (R Project for Statistical Computing, RRID:SCR_001905) [6].

The AFRICOS study was conducted in accordance with the ethical principles outlined in the Declaration of Helsinki. Ethical approval was obtained from Institutional Review Boards in each participating country: the Kenya Medical Research Institute Scientific and Ethics Review Unit (KEMRI SERU) in Kenya; the Uganda National Council for Science and Technology (UNCST) in Uganda; the National Health Research Ethics Committee of the National Institute for Medical Research (NIMR) in Tanzania; the National Health Research Ethics Committee (NHREC) in Nigeria; and the Walter Reed Army Institute of Research Institutional Review Board (WRAIR IRB) in the United States. All participants, including both adults and minors, provided written informed consent or assent prior to enrollment. For minors, assent was obtained alongside written consent from their parents or legal guardians. Participant recruitment commenced on 21 January 2013 and remains ongoing.

## Results

### Participant characteristics and HBV prevalence

Among 4,290 participants enrolled between January 2013 and November 2024, 4,241 (98.9%) had a valid HBV test at any visit, of whom 3,439 (81.1%) were living with HIV and 802 (18.9%) were not living with HIV (Fig 1). A total of 195

# CONSORT Flow Diagram

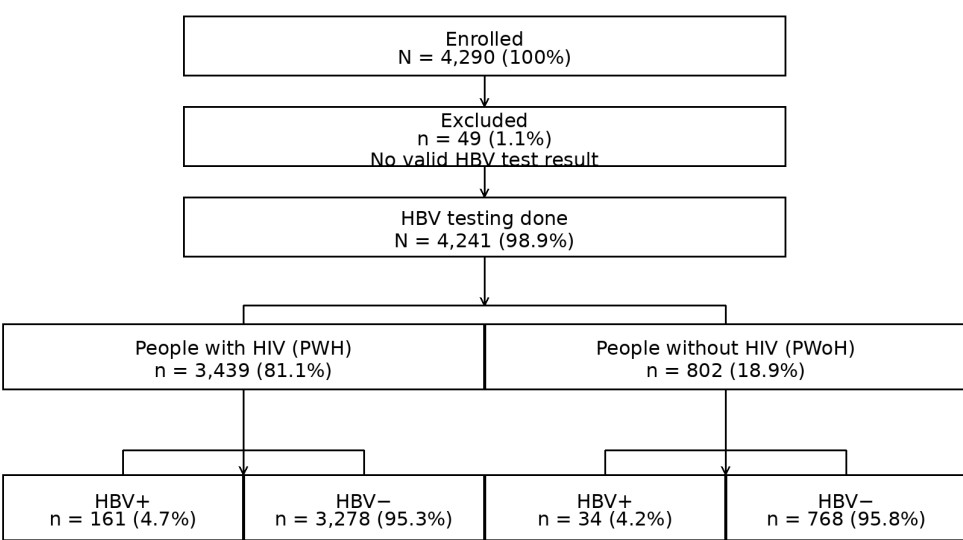

**Fig 1. Flow diagram of participant enrollment and HBV testing.** A total of 4,290 participants were enrolled; 49 (1.1%) were excluded due to missing or invalid hepatitis B virus (HBV) results. HBV testing was completed for 4,241 participants (98.9%). Of these, 3,439 (81.1%) were people with HIV (PWH) and 802 (18.9%) were people without HIV (PWoH). HBV prevalence was 4.7% (161/3,439) among PWH and 4.2% (34/802) among PWoH. Abbreviations: HBV, hepatitis B virus; PWH, people with HIV; PWoH, people without HIV.

(4.6%) were diagnosed with HBV at the visit with their first valid test. The prevalence of HBV among PLWH was 4.7% (161/3439) while it was 4.2% (34/802) among people without HIV, corresponding to a relative risk of 1.10 (95% CI: 0.77–1.59, p = 0.601).

## Hepatitis B co-infection among people living with HIV

Of the 3,439 PLWH, 609 (17.7%) were from Uganda, 661 (19.2%) from Tanzania, 389 (11.3%) from Nigeria, and 1,780 (51.8%) from Kenya (Table 1). Most PLWH in our study were female (58.0%) and most were aged 25 years and above (39.4% aged 25–39 and 37.3% aged ≥40). The prevalence of HBV co-infection among PLWH varied by country, with Nigeria having the highest prevalence (11.1%) and Kenya having the lowest (2.6%, p < 0.001; Fig 2A). The prevalence of HIV/HBV co-infection was higher among males (6.7%) as compared to females (3.2%, p < 0.001; Fig 2B). The prevalence of HIV/HBV co-infection also increased with age, from 0.2% among PLWH aged 15–19 years to 5.8% among PLWH aged 40 years or older (p < 0.001; Fig 2C).

## Factors associated with HBV co-infection among people living with HIV

Logistic regression was applied to identify factors independently associated with HBV co-infection among PLWH (Table 1). Males were noted to be at over twice the odds of HBV co-infection compared to females (OR [95% CI]: 2.22 [1.59–3.09]; p < 0.001). Participants from Nigeria had about five times higher odds (5.19 [3.35–8.05]; p < 0.001) compared to participants from Kenya, with the older age groups having significantly higher odds of co-infection (25–39 years: 2.78 [1.61–4.79]; p < 0.001; ≥40 years: 2.99 [1.74–5.13]; p < 0.001) compared to participants aged 15–24 years (Fig 3).

**Table 1. Demographic Distribution of Hepatitis B Co-infection Among People with HIV (n = 3439).**

| | Overall (N = 3439) | HBV+ (N = 161) | HBV- (N = 3279) | p-value[1] | Unadjusted OR (95% CI) | p-value | Adjusted OR (95% CI) | p-value |
|---|---|---|---|---|---|---|---|---|
| | n (col%) | n (row%) | n (row%) | | | | | |
| **Country** | | | | <0.001 | | | | |
| Uganda | 609 (17.7%) | 39 (6.4%) | 570 (93.6%) | | 2.58 (1.67, 3.99) | <0.001 | 2.78 (1.79, 4.31) | <0.001 |
| Kenya | 1780 (51.8%) | 46 (2.6%) | 1734 (97.4%) | | Reference | | | |
| Tanzania | 661 (19.2%) | 33 (5.0%) | 628 (95.0%) | | 1.98 (1.25, 3.12) | 0.003 | 2.16 (1.36, 3.42) | 0.001 |
| Nigeria | 389 (11.3%) | 43 (11.1%) | 346 (88.9%) | | 4.68 (3.04, 7.21) | <0.001 | 5.19 (3.35, 8.05) | <0.001 |
| **Sex** | | | | <0.001 | | | | |
| Male | 1445 (42.0%) | 97 (6.7%) | 1348 (93.3%) | | 2.17 (1.57, 3.00) | <0.001 | 2.22 (1.59, 3.09) | <0.001 |
| Female | 1994 (58.0%) | 64 (3.2%) | 1930 (96.8%) | | Reference | | | |
| **Age (years)** | | | | <0.001 | | | | |
| 15-19 | 416 (12.1%) | 1 (0.2%) | 415 (99.8%) | | Reference (combined 15–24 years category) | | | |
| 20-24 | 383 (11.1%) | 16 (4.2%) | 367 (95.8%) | | | | | |
| 25-39 | 1355 (39.4%) | 69 (5.1%) | 1286 (94.9%) | | 2.47 (1.44, 4.23) | <0.001 | 2.78 (1.61, 4.79) | <0.001 |
| ≥40 | 1284 (37.3%) | 75 (5.8%) | 1209 (94.2%) | | 2.85 (1.67, 4.87) | <0.001 | 2.99 (1.74, 5.13) | <0.001 |
| Missing | 1 | 0 | 1 | | | | | |

[1] p-value from chi-squared test or Fisher's exact test (for variables with any cell size <5). Missing values were excluded from these tests.

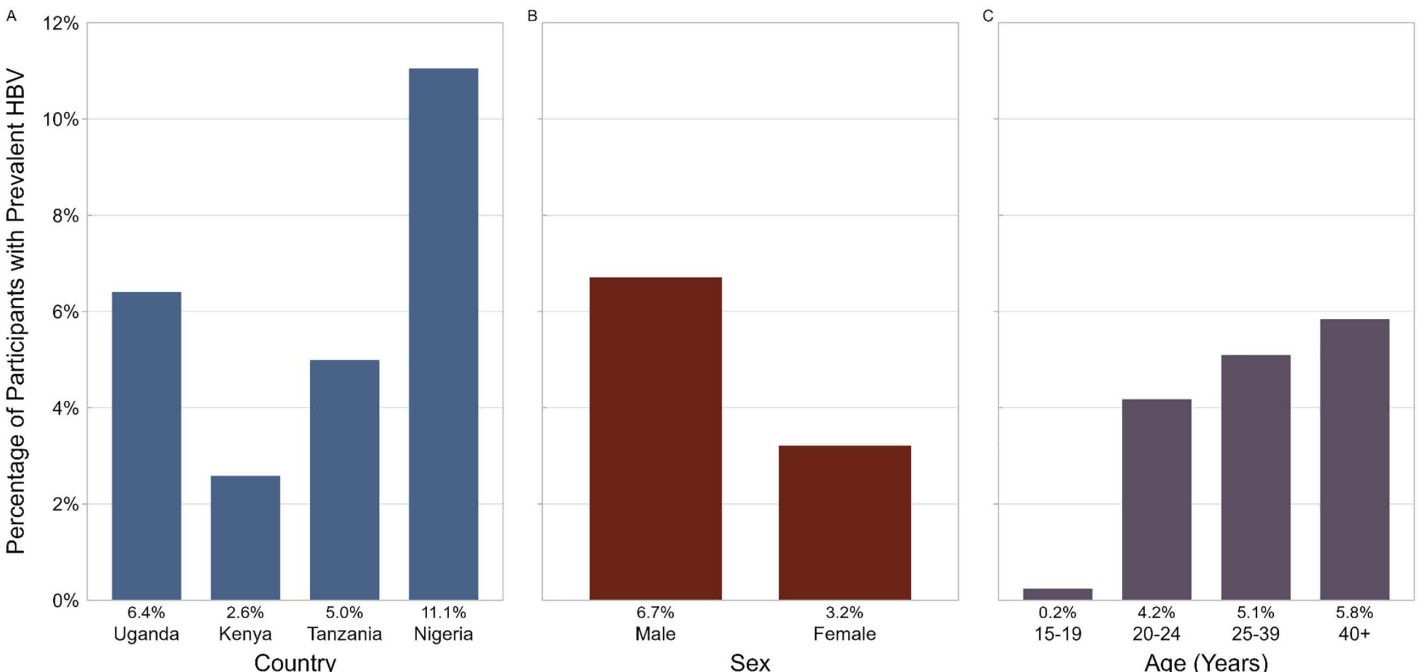

**Fig 2. Prevalence of hepatitis B virus among people with HIV by country, sex, and age group. (A)** HBV prevalence by country among people with HIV (PWH), ranging from 2.6% to 11.1%. **(B)** HBV prevalence by sex, with higher prevalence among males than females. **(C)** HBV prevalence by age group, demonstrating an increase from adolescence to older age groups. Abbreviations: HBV, hepatitis B virus; PWH, people with HIV.

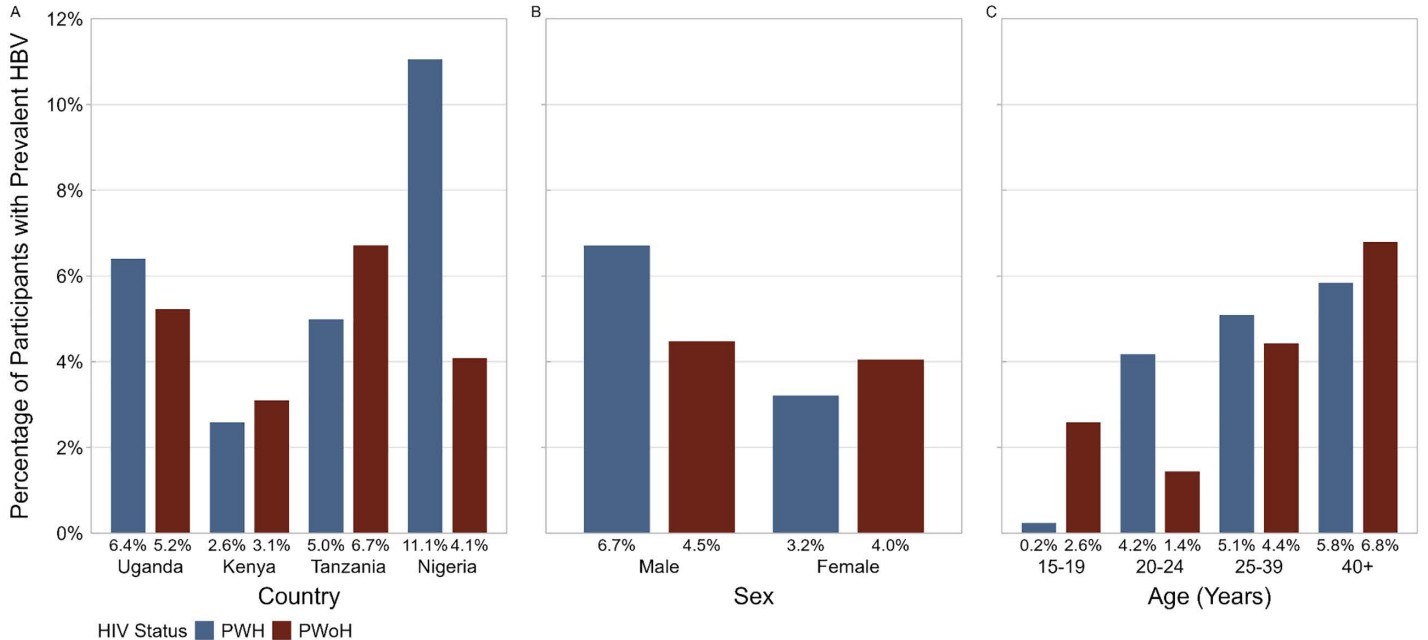

**Fig 3. Prevalence of hepatitis B virus by HIV status, country, sex, and age group. (A)** HBV prevalence by country among people with HIV (PWH) and people without HIV (PWoH). **(B)** HBV prevalence by sex, with higher prevalence among males in both groups. **(C)** HBV prevalence by age group, illustrating differing age-related patterns between PWH and PWoH. Abbreviations: HBV, hepatitis B virus; PWH, people with HIV; PWoH, people without HIV.

## Discussion

The overall HBV prevalence of 4.7% among PLWH in our cohort is slightly lower but broadly consistent with regional estimates from prior studies, which have reported co-infection rates ranging from 5% to 15% across Sub-Saharan Africa, depending on the population and diagnostic methods used [5,7,9].

Nigeria exhibited the highest co-infection prevalence (11.1%), aligning with previous research that has identified West Africa as a region with particularly high HBV endemicity [8]. In contrast, Kenya had the lowest prevalence (2.6%), consistent with prior studies indicating a lower HBV burden in East Africa [9]. This difference observed across countries likely reflects differences in regional HBV endemicity, transmission patterns, and health system factors. West Africa has historically high HBV endemicity, partly due to more common perinatal and early childhood transmission, whereas East Africa tends to have lower chronic HBV rates with more horizontal transmission later in life [8,9]. Other factors may include variability in HBV vaccination coverage, as well as differences in diagnostic practices, healthcare access, and population-level risk factors. These factors influence the burden of HBV co-infection among PLWH in different settings. Although slightly lower, our findings are consistent with studies from other African regions. Shivakumar et al. (2024) reported an HBV prevalence of 7.1% among adults living with HIV in South Africa, a region that bears a significant burden of the HIV epidemic [10]. Similarly, a systematic review from Ethiopia estimated HBV prevalence among PLWH to range from 5% to 17% [11], while a study in Senegal reported an HBV seroprevalence of 11.9% among PLWH [12]. In Europe, the EuroSIDA study found a 9% HIV/HBV co-infection prevalence, highlighting the consistent global intersection of these infections [13]. Additionally, the meta-analysis by Tengan et al., reported a pooled HBV prevalence of 7.0% among PLWH in Latin America and the Caribbean, suggesting that HBV co-infection remains a significant challenge in multiple regions globally [14]. These international

comparisons reinforce the broader applicability of our results in demonstrating the importance of routine HBV screening and integrated treatment models in HIV care both within and beyond Africa.

Males showed significantly higher odds of HBV co-infection than females, a finding supported by prior studies [15,16]. Several factors may contribute to this disparity. First, studies have shown that men are more likely to engage in behaviors associated with HBV transmission than women; multiple sexual partnerships, inconsistent condom use, and injecting drug use [17]. Second, health-seeking patterns differ with women generally more likely than men to access healthcare services (that include antenatal care) and this often leads to earlier diagnosis and treatment in women [18]. Third, biological and immunological differences between males and females may contribute to disparities. Estrogen enhances immune responses, whereas testosterone has immunosuppressive effects [19]. Females are more likely to clear HBV after acute infection, whereas males are at greater risk of developing chronic HBV, which could explain the higher observed prevalence in men [20]. Taken together, these behavioral, healthcare access, and biological differences help explain the sex-based disparities in HBV prevalence among people living with HIV.

Individuals aged 25 years and older had significantly higher odds of co-infection. This likely reflects cumulative exposure over time, particularly in regions where HBV transmission occurs both perinatally and through horizontal means in adolescence and adulthood [21]. This trend has been consistently observed in prior studies from both Africa and other global regions. For example, Ondondo et al. (2024) in Kenya reported increasing HBV prevalence with age among PLWH, likely due to cumulative exposure and low childhood vaccination coverage prior to its inclusion in Kenya's expanded program on immunization [22]. Similarly, Belyhun et al. (2016) noted a higher burden of HBV in older individuals across multiple African settings, reflecting horizontal transmission in adolescence and adulthood, especially in areas with late implementation of birth-dose HBV vaccination [5,11]. Global studies, including those by Thio et al. (2002), Shivakumar et al. (2024), and Tengan et al. (2017), further support this association between age and HBV co-infection risk among PLWH [10,14,21]. However, some studies have found less consistent age trends. For instance, Diop-Ndiaye et al. (2008) in Senegal and the EuroSIDA study (Price et al., 2013) did not identify a strong age-related pattern [12,13]. This may be due to different patterns of HBV transmission (e.g., more iatrogenic or MSM-associated in high-income countries), better vaccine uptake, or earlier screening and treatment in younger individuals. Despite some heterogeneity of findings from different studies, the overall evidence underscores the importance of targeting older adults for HBV screening and treatment in HIV programs, particularly in regions with historically low immunization coverage.

Our findings underscore the need to integrate HBV testing into routine HIV care. Historically, most standard ART regimens have included agents such as tenofovir disoproxil fumarate (TDF), which also suppress HBV replication [23,24]. As a result, HBV has often been managed incidentally in PLWH, without explicit diagnosis or monitoring. However, newer TDF-sparing regimens—such as long-acting injectables—lack HBV activity [25]. Initiating these regimens in individuals with undiagnosed chronic HBV may lead to viral reactivation and hepatitis flares due to the sudden withdrawal of HBV-suppressive agents [26].

Our findings should be interpreted with caution. HBV infection was assessed using a single serological marker (HBsAg) at one time point, which limited our ability to distinguish acute, chronic, or resolved infection. This approach may have included rare false positives from resolved infections that remain HBsAg-positive, while also missing occult HBV, particularly among individuals on ART where HBsAg detection can be suppressed. Participants were recruited from PEPFAR-funded clinical programs predominantly in urban settings, so our findings may not reflect the broader populations of people with and without HIV in each country.

HBV co-infection remains a major concern among PLWH in Sub-Saharan Africa, with variation by country, sex, and age. Scaling up HBV testing, vaccination, and integrated care is especially important in high-prevalence areas like Nigeria, where undiagnosed HBV could lead to reactivation when switching to TDF-sparing regimens. Future studies should assess long-term liver outcomes and the impact of newer ART regimens in co-infected populations.

Of 3439 people with HIV who had a valid Hepatitis B test at any visit, 161 (4.7%) had prevalent HBV at the time of their first valid test. One participant with missing demographic variables was excluded from regression analyses (n = 3438). Due to the small number of participants aged 15–19 years who had prevalent HBV, the 15–19 and 20–24 year age categories were collapsed in the regression models. Univariable logistic regression was used to estimate unadjusted odds ratios (ORs) and 95% confidence intervals (CIs) for each demographic characteristic, and multivariable logistic regression with all three demographic variables included in the model was used to estimate adjusted ORs and 95% CIs. Demographic characteristics for inclusion in the multivariable model were chosen a priori. Statistically significant associations (p < 0.05) are bolded.

## Supporting information

**S1 File. Inclusivity in global research checklist.** Completed PLOS inclusivity in global research questionnaire outlining ethical, cultural, and scientific considerations relevant to the conduct and reporting of this study, including ethical approvals, community engagement, authorship, and considerations for human and non-human subjects research. (DOCX)

## Acknowledgments

We thank the study participants, local implementing partners, and hospital leadership at Kayunga District Hospital, Kericho District Hospital, AC Litein Mission Hospital, Kapkatet District Hospital, Tenwek Mission Hospital, Kapsabet District Hospital, Nandi Hills District Hospital, Kisumu West District Hospital, Mbeya Zonal Referral Hospital, Mbeya Regional Referral Hospital, Defence Headquarters Medical Center, and the 68th Nigerian Army Reference Hospital for their invaluable contributions to this study.

The full membership of the African Cohort Study (AFRICOS) Study Group is listed below:

U.S. Military HIV Research Program Headquarters Group:

Ajay Parikh, Brennan Cebula, Glenna Schluck, Jaclyn Hern, Jillian Chambers, Kara Lombardi, Kimberly Bohince, Leigh Anne Eller, Linsey Scheibler, Mary Schmitz, Michelle Imbach, Patricia Agaba, Sean Cavanaugh, Trevor Crowell.

AFRICOS Uganda Group:

Agatha Mugagga Mukanza, Alfred Lutaaya, Anne Nakirijja, Benard Okanyakure, Betty Mwesigwa, Cate Kafeero, Christine Nabanoba, Christine Nanteza, Claire Nakazzi Bagenda, Estella Birabwa, Evelyn Najjuma, Ezra Musingye, Fred Magala, Freddie Ssentogo, Godfrey Zziwa, Grace Mirembe, Hannah Kibuuka, Harriet Nabirye, Hellen Birungi, Hilda Mutebe, Isaac Kato Kenoly, Jacqueline Namugabo, Michael Semwogerere, Michael Waiswa, Paul Wangiri, Phiona Namulondo, Prossy Naluyima, Richard Tumusiime, Ronald Ephraim Wasswa, Sylvia Namanda, Vamsi Vasireddy.

AFRICOS South Rift Valley, Kenya Group:

Aaron Ngeno, Aggrey Koech, Alice Airo, Bornes Ngetich, Brayan Langat, Christopher Ochieng, Deborah Langat, Edwin Langat, Francis Opiyo, Grace Engoke, Ibrahim Daud, Ignatius Kiptoo, Irene Metet, Isaac Tsikhutsu, Janet Ngeno, Japhet Towett, Joan Kapkiai, Jonah Mwaswai, Joshua Rotich, Josphat Kosgei, Kennedy Labosso, Leonard Cheruiyot, Linner Rotich, Lucy Korir, Mary Leelgo, Mercy Chelimo, Michael Obonyo, Mike Bii, Raphael Langat, Rither Langat, Salome Ndungu, Samuel Kiprotich, Susan Ontango, Triza Rono, Wilfred Kirui, Zeddy Bett Kesi.

AFRICOS Kisumu, Kenya Group:

Agnes Atieno, Celine Ogari, Charles Okwaro, Doris Njoroge, Elkanah Modi, George Suja, Iddah Aoko, Janet Oyieko, John Owuoth, Joseph Meyo, Kennedy Obambo, Lovet Nyawanda, Lucas Otieno, Michal Ohaga, Oscar Adimo, Paul Omolo, Solomon Otieno, Trizer Achieng, Valentine Sing'oei, Winnie Rehema.

AFRICOS Mbeya, Tanzania Group:

Dorothy Mkondoo, Eliud Myegeta, Emmanuel Bahemana, Faraja Mbwayu, Gloria David, Goodluck Kisonga, Gwamaka Mwaisanga, Happy Laiton, Janeth Likiliwike, Jaquiline Mwamwaja, John Njegite, Johnisius Msigwa, Laban Mwandumbya, Lucas Maganga, Mtasi Mwaipopo, Naima Mkingule, Paschal Kiliba, Peter Edwin, Raphael Mkinga, Reginald Gervas, Restituta Minde, Rose Bruno, Vumilia Kaduma, Willyhelmina Olomi.

AFRICOS Abuja, Nigeria Group:

Abdulwasiu Bolaji Tiamiyu, Aminu Suleiman, Blessing Edet Okon, Chisara Okolo, Felicia Anayochukwu Odo, Feyisayo Jegede, Helen Nwandu, Ifeanyi Okoye, Ijeoma Chigbu-Ukaegbu, Lawrence Umeji, Mfreke Asuquo, Ndubuisi Okeke, Onyinye Eze, Rosemary Akiga, Samirah Mohammed, Sunday Odeyemi, Victor Anyebe, Yakubu Adamu.

AFRICOS Lagos, Nigeria Group:

Abdulkadir Ramatu, Adewale Adelakun, Aire Commodore Edward Akinwale (Rtd), Blessing Irekpitan Wilson, Chiamaka Modesta Ibeanu, Concilia Uzoamaka Agbaim, Elekwa Chinenye Elizabeth, Igiri Faith, Jacinta Elemere, Jumoke Titilayo Nwalozie, Lt Col Sulaimon Awesu, Maj Christian Etim Efut, Ndubuisi Rosemary Obiageli, Nkechinyere Harrison, Nnadi Theodora Nkiru, Obende Theresa Owanza, Obilor Ifeoma Lauretta, Olutunde Ademola Adegbite, Rachael Eluwa, Uzoegwu Amaka Natalie, Victoria Idi, Yakubu Adamu, Zahra Parker.

This list represents the complete AFRICOS Study Group membership at the time of this study.

Disclaimer: This material has been reviewed by the Walter Reed Army Institute of Research. There is no objection to its presentation and/or publication. The opinions or assertions contained herein are the private views of the author, and are not to be construed as official, or as reflecting true views of the Department of the Army or the Department of Defense. The investigators have adhered to the policies for protection of human research participants as prescribed in AR 70–25.

## Author contributions

**Conceptualization:** Josphat Kosgei, Leigh Anne Eller, Julie A. Ake, Trevor A. Crowell.

**Data curation:** Josphat Kosgei, Natalie Burns.

**Formal analysis:** Josphat Kosgei, Natalie Burns, Trevor A. Crowell.

**Funding acquisition:** Julie A. Ake.

**Investigation:** Josphat Kosgei, Hannah Kibuuka, Leigh Anne Eller, Julie A. Ake, Trevor A. Crowell.

**Methodology:** Julie A. Ake.

**Project administration:** Julie A. Ake.

**Resources:** Julie A. Ake.

**Supervision:** Walter Jaoko, Julius Oyugi, Julie A. Ake, Trevor A. Crowell.

**Writing – original draft:** Josphat Kosgei.

**Writing – review & editing:** Josphat Kosgei, Walter Jaoko, Julius Oyugi, Natalie Burns, Jonah Maswai, John Owuoth, Valentine Sing'oei, Emmanuel Bahemana, Abdulwasiu B. Tiamiyu, Hannah Kibuuka, Leigh Anne Eller, Julie A. Ake, Neha Shah, Trevor A. Crowell.

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
