## [Decision Letter · Decision Letter 0]

4 Jan 2026

PONE-D-25-59536Epidemiology of Hepatitis B Virus Co-infection Among People Living with HIV in Four Countries in Sub-Saharan AfricaPLOS One

Dear Dr. Kosgei,

Thank you for submitting your manuscript to PLOS ONE. After careful consideration, we feel that it has merit but does not fully meet PLOS ONE’s publication criteria as it currently stands. Therefore, we invite you to submit a revised version of the manuscript that addresses the points raised during the review process. Specificallly please answer to the comment that came from the IA system used by the PlosOne editorial staff I received as a new tools. I'm not fully convinced by this automatic review, thus I would like you provide me your feedback on the different items they evaluated. Please submit your revised manuscript by Feb 18 2026 11:59PM. If you will need more time than this to complete your revisions, please reply to this message or contact the journal office at plosone@plos.org. Please include the following items when submitting your revised manuscript:

We look forward to receiving your revised manuscript.

Kind regards,

Pierre Roques, Ph.D.

Academic Editor

PLOS One

Journal Requirements:

4. In the online submission form, you indicated that the datasets generated and/or analyzed during the current study are available from the corresponding author on reasonable request. To request a minimal data set, please contact the Data Coordinating and Analysis Center (DCAC) at PubRequest@hivresearch.org and indicate the AFRICOS (RV329) study along with the name of the manuscript.

6. Please amend your list of authors on the manuscript to ensure that each author is linked to an affiliation. Authors’ affiliations should reflect the institution where the work was done (if authors moved subsequently, you can also list the new affiliation stating “current affiliation:….” as necessary).

7. One of the noted authors is a group or consortium [African Cohort Study (AFRICOS) Study Group]. In addition to naming the author group, please list the individual authors and affiliations within this group in the acknowledgments section of your manuscript. Please also indicate clearly a lead author for this group along with a contact email address.

8. Please include a separate caption for each figure in your manuscript.

9. Please include your tables as part of your main manuscript and remove the individual files. Please note that supplementary tables (should remain/ be uploaded) as separate "supporting information" files.

10. Please include captions for your Supporting Information files at the end of your manuscript, and update any in-text citations to match accordingly. Please see our Supporting Information guidelines for more information: http://journals.plos.org/plosone/s/supporting-information.

11. We note that there is identifying data in the Supporting Information file <RV 329a WRAIR #1897A 2025 CRR Approval.zip>. Due to the inclusion of these potentially identifying data, we have removed this file from your file inventory. Prior to sharing human research participant data, authors should consult with an ethics committee to ensure data are shared in accordance with participant consent and all applicable local laws.

-Location data

Reviewers' comments:

Reviewer's Responses to Questions

**Comments to the Author**

1. Is the manuscript technically sound, and do the data support the conclusions?

Reviewer #1: Yes

Reviewer #2: Yes

2. Has the statistical analysis been performed appropriately and rigorously?

Reviewer #1: Yes

Reviewer #2: Yes

3. Have the authors made all data underlying the findings in their manuscript fully available?

Reviewer #1: Yes

Reviewer #2: Yes

4. Is the manuscript presented in an intelligible fashion and written in standard English?

Reviewer #1: Yes

Reviewer #2: Yes

5. Review Comments to the Author

Reviewer #1: The authors present a summary of HBV prevalence (determined by HBsAg positivity) in a cohort including both HIV+ and HIV- participants in four countries. The analysis is straightforward and appears to be appropriate given the study design. HBV infection, and particular HBV/HIV co-infection, is an important public health problem in Africa, and, as the authors show, the burden varies widely between contexts, and has public health implications regarding choice of ART regimens and screening for HBV in HIV programs.

I do not have any major comments. The authors note the limitation that they did not fully characterize HBV infection, relying on a single marker for current infection. I would suggest changing 'markers' to 'marker' on line 230, as only a single marker was used, as per the methods. While some resolved infections may still be HBsAg+, my understanding is that most would not be, so I am not sure if I would include them in the limitation. On the other hand, some occult cases, especially due to ART interference with HBsAg positivity, may have been missed. This because from the methods it sounds as if patients could have been screened for HBsAg when already on ART at any time during their HIV care. I am not certain if a change is warranted here, but the authors could consider discussing in more detail the limitations of the testing approach given the context.

Reviewer #2: This is an automated report for PONE-D-25-59536. This report was solicited by the PLOS One editorial team and provided by ScreenIT.

ScreenIT is an independent group of scientists developing automated tools that analyze academic papers. A set of automated tools screened your submitted manuscript and provided the report below. Each tool was created by your academic colleagues with the goal of helping authors. The tools look for factors that are important for transparency, rigor and reproducibility, and we hope that the report might help you to improve reporting in your manuscript. Within the report you will find links to more information about the items that the tools check. These links include helpful papers, websites, or videos that explain why the item is important. While our screening tools aim to improve and maintain quality standards they may, on occasion, miss nuances specific to your study type or flag something incorrectly. Each tool has limitations that are described on the ScreenIT website. The tools screen the main file for the paper; they are not able to screen supplements stored in separate files. Please note that the Academic Editor had access to these comments while making a decision on your manuscript. The Academic Editor may ask that issues flagged in this report be addressed. If you would like to provide feedback on the ScreenIT tool, please email the team at ScreenIt@bih-charite.de. If you have questions or concerns about the review process, please contact the PLOS One office at plosone@plos.org.

6. PLOS authors have the option to publish the peer review history of their article (what does this mean?). If published, this will include your full peer review and any attached files.

Reviewer #1: No

Reviewer #2: No

---

## [Author Response · Author response to Decision Letter 1]

26 Feb 2026

The reviewers' comments were received and the responses to reviewers have been uploaded, with a line-by-line response to both Reviewer#1 and Reviewer#2 comments. Thank you for your time and reviews.

---

## [Editor Report · Decision Letter 1]

3 Mar 2026

Epidemiology of Hepatitis B Virus Co-infection Among People Living with HIV in Four Countries in Sub-Saharan Africa

PONE-D-25-59536R1

Dear Dr. Kosgei,

We’re pleased to inform you that your manuscript has been judged scientifically suitable for publication and will be formally accepted for publication once it meets all outstanding technical requirements.

Kind regards,

Pierre Roques, Ph.D.

Academic Editor

PLOS One
---

## [Editor Report · Acceptance letter]

PONE-D-25-59536R1

PLOS One

Dear Dr. Kosgei,

I'm pleased to inform you that your manuscript has been deemed suitable for publication in PLOS One. Congratulations! Your manuscript is now being handed over to our production team.

Kind regards,

on behalf of

Dr. Pierre Roques

Academic Editor

PLOS One